# Breathing Exercises for Improving Cognitive Function in Patients with Stroke

**DOI:** 10.3390/jcm11102888

**Published:** 2022-05-20

**Authors:** Eui-Soo Kang, Jang Soo Yook, Min-Seong Ha

**Affiliations:** 1Department of Sports Science Convergence-Graduate School, Dongguk University-Seoul, 30, Pildong-ro 1-gil, Jung-gu, Seoul 04620, Korea; 01062463380@naver.com; 2Center for Functional Connectomics, Brain Research Institute, Korea Institute of Science and Technology (KIST), Hwarang-ro 14-gil 5, Seongbuk-gu, Seoul 02792, Korea; soulyook84@gmail.com; 3Department of Sports Culture, College of the Arts, Dongguk University-Seoul, 30, Pildong-ro 1-gil, Jung-gu, Seoul 04620, Korea

**Keywords:** cerebrovascular disease, hemiplegia, stroke, cognitive function, breathing exercise

## Abstract

Patients with stroke may experience a certain degree of cognitive decline during the period of recovery, and a considerable number of such patients have been reported to show permanent cognitive damage. Therefore, the period of recovery and rehabilitation following stroke is critical for rapid cognitive functional improvements. As dysfunctional breathing has been reported as one of the factors affecting the quality of life post stroke, a number of studies have focused on the need for improving the breathing function in these patients. Numerous breathing exercises have been reported to enhance the respiratory, pulmonary, cognitive, and psychological functions. However, scientific evidence on the underlying mechanisms by which these exercises improve cognitive function is scattered at best. Therefore, it has been difficult to establish a protocol of breathing exercises for patients with stroke. In this review, we summarize the psychological, vascular, sleep-related, and biochemical factors influencing cognition in patients and highlight the need for breathing exercises based on existing studies. Breathing exercises are expected to contribute to improvements in cognitive function in stroke based on a diverse array of supporting evidence. With relevant follow-up studies, a protocol of breathing exercises can be developed for improving the cognitive function in patients with stroke.

## 1. Introduction

Stroke is caused by a fall in blood supply to the brain or due to cerebral hemorrhage and is the most common cerebrovascular condition [1]. With the increase in the average life expectancy as a result of lifestyle improvements and advances in health care, the number of patients with stroke has seen an upward trend [2]. Furthermore, studies have shown a possible association between the coronavirus disease (COVID-19) and acute cerebrovascular diseases such as ischemic stroke, hemorrhagic stroke, and cerebral venous thrombosis [3]. With the continuing COVID-19 pandemic, a steep increase in stroke incidence is therefore likely.

Health-related quality of life (HRQoL), comprising physical, psychological, and cognitive functions, is substantially low among stroke survivors [4,5,6]. In particular, decline in attention, memory, and executive function is a serious problem in patients who have had a stroke [7]. Indeed, as high as 80% of patients with stroke are presumed to have cognitive decline [8] with more than 1 in 3 of them showing permanent decline [9,10,11]. Since cognitive function plays a critical role in the performance of most activities of daily living [12,13], rapid cognitive functional improvement is recognized as an important component of recovery and rehabilitation after stroke.

The following mechanism has been proposed for the pathophysiological role of breathing in stroke—hemiplegia causes an asymmetry in the body, which leads to inefficient movements, with a direct or indirect impact on the activation of respiratory muscles, ultimately affecting the respiratory cycle [14]. Muscles such as the diaphragm, transverse abdominis, and pelvic floor perform local stabilization, with dual roles in breathing and trunk stabilization [15,16,17]. When stroke adversely affects the activation of those muscles, trunk stabilization is impacted [18,19], which affects the muscles related to breathing [20,21]. Such pathological progression may lead to a vicious cycle, wherein the abnormal function of the muscles related to breathing affects trunk stabilization, which in turn affects breathing in patients with stroke [22,23]. This could be one of the factors deteriorating the HRQoL in these patients [24,25].

In addition, pulmonary complications, such as respiratory failure, pneumonia, pleural effusion, acute respiratory distress syndrome, pulmonary edema, and pulmonary embolism from venous thromboembolism, are commonly observed in patients with stroke [26]. Pulmonary function has been demonstrated to be significantly decreased in patients with stroke compared to that in healthy individuals [27]. In particular, the activity of the diaphragm has been shown to be reduced in patients with stroke with hemiplegia [19], which suggests that the respiratory muscle has a critical role in the recovery of pulmonary dysfunction after stroke. Previous studies have attempted to test interventions that improve the respiratory and pulmonary function of patients with stroke [14,28,29,30]. Furthermore, a meta-analysis reported that respiratory muscle exercises enhanced muscular strength and reduced the risk of respiratory complications in patients with stroke [31]. Despite adequate levels of chest wall expansion, the tidal ventilation, lung capacity, and lung volume should be maintained to improve the efficiency of breathing in patients with stroke [32]. However, the exact mechanisms of such effects have not been sufficiently studied, with a consequent lack of a suitable protocol related to breathing exercises for stroke.

Furthermore, the specific mechanism by which enhanced breathing improves cognitive function in patients with stroke is yet to be elucidated. However, the positive effects of breathing exercises on cognitive functional improvement have been reported in studies although not on patients with stroke. Multiple studies on healthy adults have reported a positive effect of breathing exercises on cognitive functional improvement [33,34,35]. Considering the synchronization of natural breathing and neuronal activity, breathing activates the cortex, hippocampus, and amygdala that are related to memory performance [36]. Abnormal breathing, such as mouth breathing in children, has been reported to reduce academic achievement and memory [37]. These studies suggest an association between normal breathing and cognitive function. Another study on patients with stroke indicated a potential interaction between breathing and cognitive function—walking activated the cortex to increase the respiratory demand, and thus, walking ability was reduced upon engagement of the respiratory cortex [38]. Therefore, it is necessary to investigate the influence of respiratory function and breathing exercises on cognitive function in patients with stroke. Herein, we review previous studies on the factors related to cognitive function in patients with stroke and the association between those factors and breathing (Figure 1).

## 2. Method

We performed the literature review using the terms “stroke”, “breathing pattern disorder”, “stroke vs. breathing pattern disorder”, “stroke vs. cognitive decline”, “stroke vs. depression vs. cognition”, “stroke vs. psychological factor vs. cognition”, “stroke vs. vascular factor vs. cognition”, “stroke vs. hypertension vs. cognition, “stroke vs. sleep factor vs. cognition”, “stroke vs. OSA vs. cognition”, “stroke vs. biochemical factor vs. cognition”, “stroke vs. IL-6 vs. cognition”, “psychological factor vs. breathing exercise”, “depression vs. breathing exercise”, “parasympathetic nerve vs. breathing exercise”, “vascular factor vs. breathing exercise”, “hypertension vs. breathing exercise”, “chemoreflex vs. breathing exercise”. “baroreflex vs. breathing exercise”, “sleep vs. breathing exercise”, “OSA vs. breathing exercise”, “biochemical factor vs. breathing exercise”, and “IL-6 vs. breathing exercise” along with the Medical Subject Headings (MESH) terms. The reference list of the articles was carefully reviewed as a potential source of information. The search was based on *Pubmed*, *Scopus*, and *Google Scholar* engines. Selected publications were analyzed, and their synthesis was used to write the review and support the hypothesis of the relationship between breathing exercise and stroke.

## 3. Common Factors of Cognitive Decline in Patients with Stroke

### 3.1. Psychological Factors Involved in Post-Stroke Cognitive Impairment

Cognitive function decline is a common occurrence in patients with stroke [8,39]. As a result of damage to the brain, approximately 80% of these patients exhibit a decline in cognitive function, and approximately 20–60% show depression [40,41]. The correlation between cognitive decline and post-stroke depression (PSD) is well-established [42]. The most consistent finding in systematic reviews, including the study by Mansur et al., on the predictive factors for depression after stroke is the association between PSD and post-stroke cognitive decline [43,44]. As such, numerous studies are in support of the association between cognitive decline in stroke and depression [45,46]. Of late, research interests surrounding this topic have increased remarkably.

### 3.2. Vascular Factors Involved in Post-Stroke Cognitive Impairment

Acute stroke and other types of vascular disease are well-known causes of cognitive impairment [47,48,49,50]. Asymptomatic cerebrovascular diseases, including silent brain infarcts and leukoaraiosis, manifest with cognitive impairment and are related to risk factors such as hypertension and atrial fibrillation [51,52,53,54,55]. Hypertension was observed in over 60% of patients with stroke [56], and approximately 25% of patients with stroke had atrial fibrillation [57]. Indeed, Waldstein et al. [58] reported that patients with peripheral arterial disease (PAD), as a common form of peripheral vascular disease, exhibited impairment of cognitive function. These previous studies suggest that vascular factors associated with cognitive impairment have been recognized as an important consequence of stroke [59,60].

### 3.3. Sleep-Related Factors Involved in Post-Stroke Cognitive Impairment

Sleep disorders occur frequently in patients with stroke. In a previous meta-analysis, over 50% of patients with stroke had a concomitant sleep disorder [61,62]. In addition, 68% of patients with acute ischemic stroke were estimated to have sleep disorders such as insomnia [63], and approximately 44% of patients with cerebral infarction were shown to have sleep disturbances at three months post-stroke [64]. Sleep disorder has a negative impact on various aspects of neurological recovery and quality of life [65,66,67]. Indeed, sleep disorder was shown to increase the risk of stroke recurrence [68] and had an undesirable effect on the cognitive function in patients with stroke [69].

Obstructive sleep apnea (OSA) is known to be an independent risk factor for stroke and is one of the common sleep disorders occurring in these patients. Approximately 30–70% of patients with stroke also have OSA [70]. In a recent study by Aaronson et al. [69], which was conducted on 80 patients with stroke with OSA and 67 without OSA, a higher decline in cognitive function was revealed in those with OSA. These results suggest that sleep disorder is a key factor influencing cognitive decline in patients with stroke as well as in healthy individuals [71,72,73].

### 3.4. Biochemical Factors Involved in Post-Stroke Cognitive Impairment

Blood lipid concentrations and abnormal neurotransmitter levels have been reported to be related to cognitive function decline in patients with stroke. In a study conducted on patients with acute ischemic stroke, the risk of cognitive decline was directly proportionate to non-high density lipoprotein cholesterol (non-HDL-C) levels [74]. Similarly, serum glutamate concentration was reported to be a critical predictor of the increased size of the cerebral infarct [75]. Zeydan et al. showed that a key factor in the cognitive decline in patients with stroke was the loss of glutamatergic neuron synaptic function [76]. Indeed, the preservation of glutamatergic neuron synaptic function was reported to help maintain the cognitive functions after stroke and prevent dementia [77,78]. The increase in the pleiotropic cytokine, interleukin-6 (IL-6), which induces an acute inflammatory response, was also reported to be related with cognitive decline in patients with stroke [79].

## 4. Associations between the Factors Influencing Cognitive Decline and Breathing in Stroke

Breathing exercises are known to improve the physiological, psychological, and cognitive function [80,81,82,83], and are currently suggested as a supplementary treatment for stress, anxiety, depression, asthma, and chronotropism [84,85,86,87,88]. Breathing can be either voluntary or involuntary. Voluntarily breathing is controlled by a complex feedback system involving the autonomous neural network, brain stem nucleus, limbic system, cortical areas, and the neuroendocrine system [80,84]. The voluntary control of breathing has been reported to exert a positive effect on autonomic nervous system functions, including variable heart rate, expiratory flow rate, and vagal tone [89,90,91]. Three months of slow breathing exercises have been shown to cause a decline in heart rate as well as an elevated sensitivity to cardiac response to standing in healthy individuals [92]. This study thus highlights the importance of breathing control as well as the method used for controlling one’s breathing. Furthermore, this study suggests future directions for research on breathing exercise methods to improve the breathing function in patients with stroke who frequently have breathing abnormalities.

### 4.1. Psychological Factors and Breathing in Stroke

PSD is the most common neuropsychiatric complication in patients with stroke [93,94]. Studies report that approximately one in three patients with stroke exhibit PSD [44,95,96,97]. In a study on healthy older adults, breathing exercises were shown to have a positive effect on psychological functions [98,99,100]. In a study by Brown and Gerbarg, breathing exercises were suggested to be beneficial for patients with PSD [80]. Breathing exercises restore the normal state of the autonomic system by regulating the movement of the respiratory system [92,101]. Moreover, enhanced parasympathetic nerve activity may lead to improvements in psychological as well as cognitive functions [101,102]. As the most significant positive effect of breathing exercises, enhanced parasympathetic nerve activity was shown to reduce the response to psychological stress and, consequently, exert positive effects on various domains across psychology, cognition, and behavior [80,84] (Table 1).

### 4.2. Vascular Factors and Breathing in Stroke

Stroke-induced damage to areas of the brain associated with autonomous function has a substantial influence on the blood pressure control and cardiac function during the period of recovery [103,104]. In addition, patients showing post-stroke hemiplegia exhibited a markedly low level of residual blood flow in the paretic lower limb [105,106,107]. Such problems of vascular function can negatively affect the performance of activities of daily living (ADL) and the quality of life [108,109]. Reduced physical activity can in turn affect the blood flow velocity, endothelial function, and arterial diameter through secondary reduction in blood flow [108,110].

No study has yet directly investigated the effects of breathing exercise and respiratory function on vascular function in patients with stroke; however, several studies have been conducted on patients with hypertension, which is known to be the most serious risk factor for stroke incidence.

Breathing exercises are widely acknowledged as a non-pharmaceutical intervention for the control of hypertension, a risk factor of stroke [111,112,113]. The mechanism of action is as follows: the pressor receptor stimulating the autonomic nervous system during prolonged inhalation and exhalation increases the baroreflex sensitivity (BRS) and decreases the sympathetic activity and chemoreflex activation [114,115]. In numerous studies, slow breathing exercises have shown positive effects on BRS, blood pressure, and autonomic nervous system function [114,116,117]. Hypertension is a particularly important risk factor for hemorrhagic stroke although it contributes to atherosclerotic disease that can lead to ischemic stroke as well, increasing the risk of stroke by approximately 2.87 times [118,119]. The prevalence of stroke in hypertension patients aged 50 years or above was 20% of the total population, and the prevalence continuously increased with increasing age [120]. Based on the correlation between hypertension and stroke, further studies should be conducted to determine the effects of breathing exercises on the vascular function, blood pressure, and autonomic nervous system in patients with stroke (Table 2).

### 4.3. Sleep and Breathing in Stroke

Sleep-related breathing disorders occur in more than half of patients with stroke [61,121]. They are also an independent risk factor for stroke [122,123,124] while being responsible for the risk of stroke recurrence, mortality, and deterioration of cognitive function [125,126]. Sleep-related breathing disorders include habitual snoring, upper respiratory tract resistance syndrome, aperiodic breathing, and sleep apnea syndrome. OSA refers to the partial or complete collapse of the upper airway during sleep, resulting in reduced or absent (or apnea) airflow lasting for 10 s [123]. Many OSA patients are highly likely to show cardiovascular or cerebrovascular diseases, as OSA is also associated with hypertension, a direct risk factor of stroke [120,127,128,129]. OSA is also associated with fibrinogen levels, a key independent risk factor for myocardial infarction and vascular diseases [130,131]. Elevated fibrinogen levels are correlated with increased risk of cardiovascular events in patients with stroke [130,131,132]. As OSA reduces the cerebral blood volume and decreases the blood supply to the brain, it is viewed as a risk factor for stroke [133].

In a study by Yaggi et al. conducted on 1022 adults with no history of stroke or myocardial infarction, the risk of stroke was shown to increase as the severity of OSA increased [134]. This result was verified in further follow-up studies and meta-analyses, where adults with OSA showed approximately a two-fold higher risk of stroke [135,136,137,138,139]. Additionally, untreated OSA after acute stroke increases long-term mortality and neurological outcomes [125,126], which highlights the importance of rapid treatment of post-stroke OSA.

The most well-known treatment of OSA in patients with stroke is continuous positive airway pressure (CPAP). While CPAP was shown to have positive effects on neuronal recovery, sleep, depression, and long-term survival [140,141,142,143], patients with stroke show a reduced long-term compliance to CPAP compared to healthy individuals [144]. This may be related to the difficulty in wearing and retaining the CPAP mask due to weak upper extremity and face as well as due to depression [144]. CPAP also has a role in causing phobia related to rhinocleisis [145], the arousal of the respiratory tract related to oral exposure and drying of mucous membranes [146]; therefore, there are challenges in its successful application in patients with stroke. This suggests the need for simple interventions or treatment strategies for OSA.

Among the interventions to improve OSA are a diversity of breathing re-education (BRE) approaches. These include the Buteyko method, inhalation resistance breathing training, and diaphragmatic breathing [147,148,149,150]. The BRE approach aims to improve the abnormal breathing pattern in patients with chronic hyperventilation. It involves exercises such as breath-holding and controlled breathing to restore the normal nasal/diaphragmatic pattern and treat dysfunctional breathing habits, such as abnormal mouth breathing and abnormal apical breathing or upper chest breathing [151].

Mouth breathing is associated with the severity of OSA [152,153], as it plays a part in snoring, OSA, apnea, and hypopnea [153]. In addition, patients with OSA were reported to show reduced strength of the diaphragm and muscles related to breathing compared with age- and sex-matched controls [147]. In a study by Courtney, the magnitude and stability of respiratory motor output for muscles of the upper airway was reported to be a key contributing factor across all forms of sleep apnea [147].

The BRE approach of Mckeown, which applies the Buteyko method, includes the conversion of mouth breathing to nasal breathing during rest, exercise, and sleep [151]. Such breathing exercises play a part in restoring nasal breathing, enhancing the function of the diaphragm, reducing breathing rate, and increasing the tolerance to changes in arterial carbon dioxide pressure [154]. Recent studies have shown that the BRE approach improves breathing patterns and can be beneficial for OSA patients [147,155]. However, no study has investigated the effects of breathing exercises or the re-education approaches for OSA and other sleep-related disorders in patients with stroke, suggesting the need for relevant further studies (Table 3).

### 4.4. Biochemical Factors and Breathing in Stroke

The magnitude of cerebral infarction on CT increases with the level of IL-6. IL-6 is the most well-known biochemical marker of stroke and is associated with a poor 3-month and 12-month prognosis [157,158]. Respiratory failure caused by the SARS-CoV-2 virus or COVID-19 infection is correlated with the cytokine release syndrome that leads to the need for mechanical ventilation [159]. IL-6 is known as the main chemokine inducing the cytokine-release syndrome [159,160,161]. Moreover, COVID-19 severity was shown to be associated with the risk of acute stroke [162,163], suggesting that studies on the relationship between the level of IL-6 and breathing in patients with stroke are warranted.

In a recent study using a mouse model, the neurological damage in mice with lung damage could be reduced by suppressing levels of IL-6 [164]. Although only a few studies have reported on the relationship between breathing exercises and IL-6 in patients with stroke, several studies have been conducted on different sets of participants. In a study conducted on overweight or obese adult males, the practice of breathing exercises was reported to be associated with significantly reduced levels of plasma IL-6 [165]. Similarly, in another study conducted on healthy individuals, breathing exercises were reported to have reduced the levels of IL-6 [166].

In a study conducted on healthy mammals, the respiratory system was markedly affected by an increase in IL-6 levels, which increased the mechanical ventilation during exhalation [167]. These results suggest a possible correlation between the levels of IL-6 and breathing based on its role in the pathogenesis of respiratory diseases (Table 4).

## 5. Rationale for the Development of Breathing Exercise Protocols to Attenuate Cognitive Decline in Patients with Stroke

Breathing exercises can restore the normal state of the autonomic nervous system [91,101] and increase the parasympathetic nervous system activity with the anticipated psychological changes and improved cognitive functions [101,102]. However, further research into the influence of breathing exercises on the vascular function, blood pressure, and autonomic nervous system changes in patients with stroke is necessary.

The importance of rapid treatment for post-stroke OSA has been recognized, and studies have reported on the positive effect of BRE based on the Buteyko method on breathing patterns in OSA patients [147,155]. However, no study has yet reported on the effects of general breathing exercises and BRE on OSA or other sleep-related breathing disorders in patients with stroke, and this is another gap in the literature that ought to be addressed.

Through literature review, we examined the factors related to cognitive decline in patients with stroke and their associations with normal natural breathing and breathing exercises. Breathing exercises were found to contribute to cognitive functional improvement in patients with stroke. In particular, it can be inferred that slow breathing exercise is the most helpful breathing exercise for improving cognitive function in stroke patients. However, there is a limitation in explaining the time, period, and intensity for breathing exercises; accordingly, the development of breathing exercise protocols to improve cognitive functions in patients with stroke is necessary.

The global COVID-19 pandemic that began in early 2020 has prompted focused efforts on the development of pulmonary rehabilitation using breathing exercise programs worldwide [168]. A similar high-quality breathing exercise protocol to suit the characteristics of patients with stroke would be useful in the current era. Finally, studies on the correlation between the effects of such protocols and cognitive decline that is most detected in patients with stroke would be highly beneficial.

## Figures and Tables

**Figure 1 jcm-11-02888-f001:**
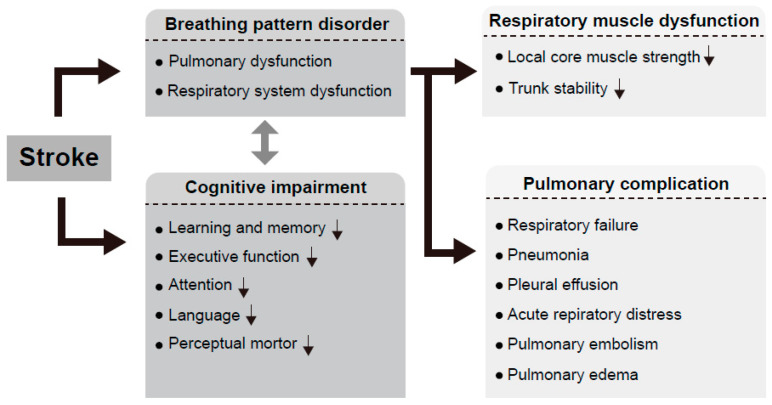
Relationship between cognitive function and breathing disorders in patients with stroke. Breathing pattern disorder and pulmonary dysfunction appearing after stroke not only cause various pulmonary complications but also affect the function of local core muscles directly related to breathing. Breathing pattern disorder has a negative effect on trunk stability and is thought to be related to cognitive decline in patients with stroke.

**Table 1 jcm-11-02888-t001:** Effects of breathing exercises on psychological factors.

Author (Year)	Participants	Time and Duration	Type	Primary Outcome	Results
Subbalakshmi et al. (2014) [91]	Healthy adults, males and females	20 min, 1 time	Acute, Nadi-shoidahana Pranayama	Basal heart rate, systolic blood pressure, peak expiratory flow rate, respiratory, cardiovascular parameters	Reduced basal heart rate and systolic blood pressure, improved peak expiratory flow rate, no difference in respiratory and cardiovascular parameters
Udupa et al. (2003) [90]	Normal young adults, males and females	20 min/d, 3 months	Long-term, Pranayama breathing	Parasympathetic and sympathetic activity	Decreased sympathetic activity, increased parasympathetic activity
Klainin-Yobas et al. (2015) [100]	Normal young adults, males and females	30 min 2 times/d, 3 months	Long-term, slow breathing	Autonomic functions	Improved autonomic functions
Hyun et al. (2009) [98]	Older adults, males and females	60 min/d, 3 times a week, 12 weeks	Long-term, Danjeon breathing	Vital capacity, physical fitness, anxiety, depression	Reduced anxiety and depression, no difference in vital capacity and physical fitness
Krishnamurthy and Telles (2007) [99]	Older adults, males and females	75 min/d, 6 times a week, 24 weeks	Long-term, Pranayama breathing with yoga training	Depression	Reduced depression

**Table 2 jcm-11-02888-t002:** Effects of breathing exercises on vascular factors.

Author (Year)	Participants	Time and Duration	Type	Primary Outcome	Results
Kalaivani et al. (2019) [111]	Hypertension patients, males and females	10 min 2 times/d, 5 days	Short-term, alternate nostril breathing	Hypertension	Reduced hypertension
Mourya et al. (2009) [112]	Hypertension patients, males and females	15 min 2 times/d, 3 months	Long-term, slow breathing	Hypertension, sympathetic and parasympathetic reactivity	Reduced hypertension, improved sympathetic and parasympathetic reactivity
Kaushik et al. (2006) [113]	Hypertension patients, males and females	10 min, 1 time	Acute, slow breathing	Hypertension	Reduced hypertension
Joseph et al. (2005) [114]	Hypertension patients, males and females	2 min of controlled breathing at 6 cycles/min, 1 time	Acute, slow breathing	Hypertension, baroreflex sensitivity	Reduced hypertension, enhanced baroreflex sensitivity
Bernardis et al. (2001) [115]	Healthy adults, males and females	10–15 min, 1 time	Acute, slow breathing	Hypoxic and hypercapnic chemoreflex, baroreflex sensitivity	Reduced chemoreflex, enhanced baroreflex sensitivity
Kalaivani et al. (2019) [111]	Hypertension patients, males and females	10 min 2 times/d, 5 days	Alternate nostril breathing	Hypertension	Reduced hypertension

**Table 3 jcm-11-02888-t003:** Effects of breathing exercises on sleep factors.

Author (Year)	Participants	Time and Duration	Type	Primary Outcome	Results
Ojay and Ernst (2000) [156]	Chronic snorers	20 min/d, 3 months	Long-term, diaphragmatic breathing and singing and exercises training	Snoring, nasal problem	Reduced snoring
Vranish and Bailey (2016) [148]	OSA patients	5 min/d, 6 weeks	Long-term, inspiratory muscle training	Respiratory muscle strength, sleep, snoring, inflammation, metabolism	Improved respiratory muscle strength and improved sleep, reduced inflammation, improved metabolism
Birch (2021) [149]	Practitioners and OSA patients	15 min 3 times/d, 2 weeks	Short-term, breathing retraining (Buteyko berating exercises)	Sleep, breathing pattern, general health, quality of life	Improved sleep, improved breathing pattern, improved general health, improved quality of life
Birch (2004) [150]	44-year-old male (with asthma, severe COPD, OSA) (Case study *n* = 1)	15 min 3 times/d, 2 years	Long-term breathing retraining (Buteyko berating exercises)	CPAP, OSA	Improved CPAP, improved OSA

**Table 4 jcm-11-02888-t004:** Effects of breathing exercises on biochemistry factors.

Author (Year)	Participants	Time and Duration	Type	Primary Outcome	Results
Sudarku (2010) [166]	University students, males and females	4 set 3 times/d, 7 weeks	Long-term, slow breathing	IL-6, IL-4, IL-2, cortisol, beta endorphin, IgG	Decreased IL-6, decreased IL-4, increased beta endorphin
Sparrow et al. (2021) [164]	Adult mice (mechanical ventilation-induced lung injury)		Mice were treated with anti-IL-6 antibody and anti-IL-6 receptor antibody	Neuronal injury, stress, inflammation	Reduced neuronal injury, reduced stress, reduced inflammation
Sarvottam et al. (2012) [165]	Overweight and obese males	50 min/d, 10 days	Short-term Pranayama breathing, Asanas (yoga postures)	CVD risk, IL-6, adiponectin, endotheline-1	Reduced risk of CVD, decreased IL-6, increased adiponectin

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
