# Peer review of "Breathing Exercises for Improving Cognitive Function in Patients with Stroke"

_jcm, 2022, doi:10.3390/jcm11102888_

Round 1
Reviewer 1 Report
This work has an original objective and a meaningful content. I congratulate the authors for the work done. I am grateful with the editors for the possibility of revising this manuscript. Although the quality of the manuscript is high, I would like to make some contributions that I hope will increase it and improve readers' understanding.
The introduction is clear and well worked.
Regarding the criteria for searching and selecting articles related to the topic of the work, it is not very clear how said search has been carried out, keywords used for the review, search string, inclusion and exclusion criteria, databases. consulted..... there are many issues that are not explained when carrying out said review and that I believe should be duly described as it is a review.
Discussion is well oriented.
Author Response
Response to Reviewers
Manuscript Title: Breathing Exercises for Improving Cognitive Function in Patients with Stroke
Editorial Board Member,
Journal of Clinical Medicine
We appreciate the constructive comments of the reviewers. We are returning herewith the above manuscript, which has been thoroughly revised as per your letter. We hope that the revised manuscript will be accepted for publication in Journal of Clinical Medicine.
To Reviewers:
We would like to thank the two reviewers for their critical comments and suggestions for our manuscript. We have read each reviewer’s comments with great care and revised the manuscript accordingly. Below we address each comment point by point. Thus, we believe that the revised version of our manuscript has been substantially improved compared to the initial submission.
Reviewer 1
This work has an original objective and a meaningful content. I congratulate the authors for the work done. I am grateful with the editors for the possibility of revising this manuscript. Although the quality of the manuscript is high, I would like to make some contributions that I hope will increase it and improve readers' understanding.
The introduction is clear and well worked.
Regarding the criteria for searching and selecting articles related to the topic of the work, it is not very clear how said search has been carried out, keywords used for the review, search string, inclusion and exclusion criteria, databases. consulted..... there are many issues that are not explained when carrying out said review and that I believe should be duly described as it is a review.
Discussion is well oriented.
Thank you for your kind and helpful review. We tried to apply your comments thoughtfully. And English language editing has been conducted by a native English speaker. We look forward to your positive evaluation.
Author’s response: We would like to thank the reviewer for their kind comments. We will try to explain in detail what you point out and we took a closer look again to confirm our error. Based on your points, we have added a method section. We highlight the corrections in red.

Reviewer 2 Report
The authors encapsulate the psychological, vascular, sleep-related as well as biochemical parameters affecting cognition in stroke patients and demonstrated the need for breathing exercise concerning the existing literature. It was postulated from the review that breathing exercises can be devised to enhance the cognitive function of patients with stroke. Overall, the paper is interesting, and the review may be beneficial to the stakeholders in this domain. Nonetheless, certain issues need to be addressed to improve the quality of the paper.
- The paper lacks proper structure based on literature selection. The authors ought to detail the procedures of literature selection criteria (e.g., Years, keywords used, literature including and excluding criteria). In other words, the paper does not follow the standard protocol for writing a review paper.
- The authors should at least provide some recommendations for the breathing exercise that might be likely beneficial to the patients.
- L113-114-related due to…Please check usage
Author Response
Response to Reviewers
Manuscript Title: Breathing Exercises for Improving Cognitive Function in Patients with Stroke
Editorial Board Member,
Journal of Clinical Medicine
We appreciate the constructive comments of the reviewers. We are returning herewith the above manuscript, which has been thoroughly revised as per your letter. We hope that the revised manuscript will be accepted for publication in Journal of Clinical Medicine.
To Reviewers:
We would like to thank the two reviewers for their critical comments and suggestions for our manuscript. We have read each reviewer’s comments with great care and revised the manuscript accordingly. Below we address each comment point by point. Thus, we believe that the revised version of our manuscript has been substantially improved compared to the initial submission.
Reviewer 2
The authors encapsulate the psychological, vascular, sleep-related as well as biochemical parameters affecting cognition in stroke patients and demonstrated the need for breathing exercise concerning the existing literature. It was postulated from the review that breathing exercises can be devised to enhance the cognitive function of patients with stroke. Overall, the paper is interesting, and the review may be beneficial to the stakeholders in this domain. Nonetheless, certain issues need to be addressed to improve the quality of the paper.
Thank you very much for your kind comment. We did our best to revise our manuscript according to your comments, and English language editing has been conducted by a native English speaker. We look forward to your positive evaluation.
Comment 1: The paper lacks proper structure based on literature selection. The authors ought to detail the procedures of literature selection criteria (e.g., Years, keywords used, literature including and excluding criteria). In other words, the paper does not follow the standard protocol for writing a review paper.
Author’s response: We appreciate your important comments. Based on your points, we have added a method section. We highlight the corrections in red.
Comment 2: The authors should at least provide some recommendations for the breathing exercise that might be likely beneficial to the patients.
Author’s response: The sharp points of reviewers make our research papers more valuable. We have carefully revised and supplemented them in the section 5 (Rationale for the development of breathing exercise protocols to attenuate cognitive decline in patients with stroke) to support the power of our research. We highlight the corrections in red.
Comment 3: L113-114-related due to…Please check usage
Author’s response: Thanks for pointing it out. We've modified that sentence to make it smoother.

Round 2
Reviewer 1 Report
The manuscript has been sufficiently improved.
Author Response
Thank you for your kind comments. We did our best to complete our review paper.
Thank you very much.

Reviewer 2 Report
I thank the authors for addressing all the comments and concerns raised.
Author Response

(The authors gave the same response as above.)
